# Quantifying the Impact of Urban Form and Socio-Economic Development on China’s Carbon Emissions

**DOI:** 10.3390/ijerph19052976

**Published:** 2022-03-03

**Authors:** Cheng Huang, Yang Qu, Lingfang Huang, Xing Meng, Yulong Chen, Ping Pan

**Affiliations:** 1School of Forestry, Jiangxi Agricultural University, Nanchang 330045, China; panping@jxau.edu.cn; 2Chongqing Key Laboratory of Precision Optics, Chongqing Institute of East China Normal University, Chongqing 401120, China; 3Institute for Global Innovation and Development, East China Normal University, Shanghai 200062, China; 4School of Urban & Regional Science, East China Normal University, Shanghai 200241, China; 5School of Water Resources & Environmental Engineering, East China University of Technology, Nanchang 330127, China; 2019210704@ecut.edu.cn; 6Key Lab of Geographic Information Science (Ministry of Education), School of Geographic Sciences, East China Normal University, Shanghai 200241, China; xmeng@geo.ecnu.edu.cn; 7Key Research Institute of Yellow River Civilization and Sustainable Development & Collaborative Innovation Center on Yellow River Civilization of Henan Province, Henan University, Kaifeng 475001, China; 52153901016@stu.ecnu.edu.cn

**Keywords:** Spatial-LMDI, carbon emissions, driving forces, urban form

## Abstract

Carbon emissions (CEs) are one of the most important factors causing global warming. The development of social economy and the acceleration of the urbanization process leads to increasing CEs, especially in China. Therefore, it has become an international community consensus to control the growth of CEs and mitigate global warming. Understanding the changing patterns and driving forces of CEs are important prerequisites for formulating policies that target the reduction of CEs in response to global warming. This study developed an improved logarithmic mean Divisia index (Spatial-LMDI) to explore the urban form and socio-economic driving forces of CEs in China. Comparing with previous studies, this study is unique in the way of applying spatial landscape index to LMDI decomposition analysis. The results show that population, per capita GDP, investment intensity and urban expansion are the top driving forces of CEs growth from 1995 to 2019. Investment efficiency, technology level, and aggregation are the most important factors in terms of restraining the growth of CEs. To achieve the goal of energy saving, CEs reduction and climate change mitigation, we proposed that strategies should be formulated as follows: improving efficiency of energy investment, optimizing the spatial distribution of construction land aggregation, and rationalizing distribution of industries.

## 1. Introduction

Global warming is one of the most serious threats humankind faces, and one major cause is the carbon dioxide emissions [1]. Carbon dioxide emission comes mainly from urban areas [2]. Studies have shown that urban areas, accounting for 2% of the global land area, generated 80% of greenhouse gas emissions, 70% of which is carbon dioxide [3,4]. Fossil energy consumption is one major source of carbon dioxide [5], and China has become the world’s largest consumer of fossil energy, as well as the largest emitter of carbon dioxide [6,7]. Data show that China’s carbon emissions (CEs) accounted for 28.17% of the global total in 2015, reaching 9.12 billion tons [8,9]. With China’s growing economy, the energy consumption and CEs keep increasing proportionately, thus China is facing increasingly severe energy supply and environmental problems [10,11]. Effective actions are urgently required for government and society to effectively reduce the growing CEs under global warming, so as to improve the living environment and achieve sustainable development.

In response to global warming, the international communities have come to scientific consensus and brought out some solutions. Major countries have signed the Kyoto Protocol and the Paris Agreement under the United Nations Framework Convention on Climate Change (UNFCCC), focusing on reducing CEs and achieving sustainable development from socio-economic perspective [12,13]. The Chinese government has actively participated in global climate change governance and international cooperation, and signed the Kyoto Protocol in 1998, and then the Paris Agreement in 2015 [14], demonstrating that China is taking a serious attitude and action in terms of mitigating global climate change. In addition, the Chinese government has formulated a quantitative greenhouse gas emission reduction target [15]. To formulate effective CEs reduction policies/plans and achieve the CEs reduction targets, it is an important prerequisite to understanding the internal driving forces of CEs growth. Quantitative research on CEs change and its influencing factors can solidify the decision-making basis for energy-saving and CEs reduction policies, and provide a reference for achieving sustainable socio-economic development.

The traditional regression method requires time-series land use data, but its availability is limited and it is unable to analyze the space form effect to CEs. The traditional driving forces analysis methods also contain a residual. Here, we propose to use the logarithmic mean Divisia index (LMDI), which contains no residuals, and time-series data is not required. LMDI model has been widely used in driving force research, and it has also been adopted for the analysis of CEs driving force [16]. Existing literature using this method focuses on global, regional, national and urban scale, respectively (Table 1). By analyzing the existing literature, it is not difficult to find that decomposition factors related to socioeconomic aspect, energy and CEs are widely used in LMDI analysis of previous studies. However, spatial factors (such as landscape and shape, area, etc.) have not been widely considered [17]. This may be due to the wide availability to socioeconomic data, and the mature and simple method to construct LMDI models using socioeconomic data. On the other hand, spatial data is relatively difficult to acquire at national level and lacks continuous data source. Therefore, the LMDI models in previous literature give a comprehensive explanation on the socio-economic, energy and CEs related driving forces of CEs, but lacking the analysis of the impact from spatial factors on CEs, especially the urban form. Moreover, Hoekstra proposed the index decomposition analysis (IDA), which is more flexible, and it only needs the aggregated data [18]. However, the practical research of this model is not enough for spatial form analysis.

In this study, we considered factors including investment efficiency (GI), fixed asset investment intensity (UInv) and other socio-economic factors, GDP and Population (P). More importantly, as an innovative methodology, spatial factors such as Splitting index (SPLIT), Effective mesh size (MESH), Edge Density (ED) and perimeter index (TE) were included in the model, these factors were urban form index. In addition, we also modified the existing LMDI model, which was named Spatial-LMDI, to explore the impact of spatial and socio-economic factors on CEs. Compared with the traditional LMDI method, the Spatial-LMDI method can effectively detect the impact of urban form on CEs. The study focused on the following aspect: Firstly, we accounted the CEs according to fossil energy consumption in China’s built-up areas at provincial-level. Among them, we mainly accounted the CEs generated by terminal energy consumption in five sectors: housing (residents’ living), industry, construction, transportation and agriculture. Secondly, through decomposition analysis, we quantitatively assessed the impact of social and economic development, built-up area expansion and urban form on CEs. The traditional LMDI model was improved, to detect the urban form driving factors, which include social–economic and urban form. Finally, we provided support and recommendations for policy formulation and climate change mitigation. This study has made contributions in three aspects. Firstly, according to the validity of the data, the time span of the sample data (from 1995 to 2019) is longer than that of the previous studies. Secondly, the sample data of CEs in this study include five sectors: industry, construction, transportation, residents’ living and agriculture, which are more comprehensive and detailed than those covered by existing research. Thirdly, this study employs GI, UInv, SPLIT, MESH, ED and TE to develop the Spatial-LMDI model, which can detect not only the traditional socio-economic driving factors of CEs change, including per capita GDP (PGDP), population (P), carbon intensity per unit GDP (UC), but also the urban form factors (such as SPLIT, MESH, ED and TE). This study explored the root causes of CEs change to provide a reference for policy-making, dealing with CEs reduction under global climate change.

## 2. Materials and Methods

### 2.1. Study Area and Materials

This study focuses on mainland China as the research area. The data of end energy consumption (physical quantity) for CEs accounting in this study were obtained from China Statistical Yearbook (CSY), China Energy Statistical Yearbook (CESY) and the National Data Website of the National Bureau of Statistics of the People’s Republic of China (http://data.stats.gov.cn/index.htm, accessed on 10 January 2021). These energy consumption data were collected on a provincial basis with 14 kinds of energy consumption, including raw coal, coal washing, coke, coke oven gas, crude oil, gasoline, diesel, kerosene, fuel oil, liquefied petroleum gas, refinery dry gas, coke oven gas, other coal washing and natural gas. Data on fixed asset investment, population and gross domestic product (GDP) were obtained from the CSY. Based on data availability, all statistical data were collected from 1995 to 2019. Land use data of China, including cultivated land, woodland, grassland, water area, construction land and unused land, were obtained from Resource and Environment Data Cloud Platform (REDCP, http://www.resdc.cn/, accessed on 10 January 2021) with a spatial resolution of 1 km. The land use data is 1995, 2000, 2005, 2010, 2015, and 2020. This study used land use of 2020 instead of 2019 land use. Provincial and national administrative boundary vector dataset were obtained from the National Geographic Information Bureau of China, which was collected and organized in 2015.

### 2.2. Methods

This study utilized a bottom-up method to quantitatively account CEs from energy consumption. This study considered the energy CEs mainly in built-up areas, so that the scenario includes the following categories: end energy consumption CEs in industrial production (Industry), residents’ living (Residential), construction (Construction), transportation (Traffic) and agriculture (Agriculture). Among them, Residential refers to the CEs from the terminal energy consumed by residents. Detailed physical consumption statistics of energy consumption departments are provided in the energy balance tables of China Statistical Yearbook (CSY) and China Energy Statistical Yearbook (CESY). The CEs accounting model is shown in Formula (1), in which the energy low calorific value data refers to China’s General Principles for Computing Comprehensive Energy Consumption, and the carbon content per unit calorific value and the oxidation rate of energy combustion data are derived from China’s Provincial Guidelines for the Compilation of Greenhouse Gas Inventories. Based on the carbon content and oxidation rate parameters, combined with General Principles for Comprehensive Energy Consumption Calculation, the parameters that can be used to calculate the carbon emissions of various types of energy in China are obtained.
(1)$$E=\sum_{j=1}^{n}C_{j}\times I_{j}=\sum_{j=1}^{n}C_{j}\times L_{j}\times P_{j}\times O_{j}\times \frac{44}{12}$$
where, *C* represents the physical quantity of energy consumption, *I* is the CEs coefficient, *J* represents the energy type, *L* is the low calorific value of energy (kJ/kg or kJ/m^3^), *P* represents the carbon content per unit calorific value, *O* represents the oxidation rate of energy combustion (%) and 44/12 represents the conversion coefficient of carbon to carbon dioxide.

#### Analysis of Driving Force

In order to better understand the impact of socio-economic development and landscape pattern changes of built-up area on CEs in the process of urbanization, we employed Spatial-LMDI model to decompose CEs into the following nine factors:(2)$$\begin{array}{l} C=P\times \frac{GDP}{P}\times \frac{Inv}{GDP}\times \frac{C}{GDP}\times \frac{GDP}{Inv}\times \frac{A^{2}}{\sum_{j=1}^{n}a_{j}^{2}}\times \frac{\sum_{j=1}^{n}a_{j}^{2}}{A}\times \frac{Ed}{A}\times \frac{1}{Ed}\\ =P\times PGDP\times UInv\times UC\times GI\times SPLIT\times MESH\times ED\times \left(TE\right) \end{array}$$
where, *C* is CEs; *P* is population; *GDP* is gross domestic product (GDP); *Inv* is fixed assets investment; *A* is area; *a_j_* is patch area of built-up areas; and *n* represents total patches of built-up areas; *Ed* is edge density. For the convenience of description, we use PGDP, UInv, UC, GI, SPLIT, MESH, ED and TE to instead $\frac{GDP}{P}$, $\frac{Inv}{GDP}$, $\frac{C}{GDP}$, $\frac{GDP}{Inv}$, $\frac{A^{2}}{\sum_{j=1}^{n}a_{j}^{2}}$, $\frac{\sum_{j=1}^{n}a_{j}^{2}}{A}$, $\frac{Ed}{A}$, $\frac{1}{Ed}$, respectively. Additionally, obtain Formula (2). *PGDP* stands for GDP per capita; *UInv* stands for investment intensity of fixed assets; *UC* stands for CEs per unit GDP; and *GI* stands for investment efficiency. *SPLIT* stands for landscape separation index; *MESH* stands for effective grain size area index; and *TE* stands for perimeter index, that is, reciprocal of edge length. SPLIT, MESH, Ed, and A were calculated by Fragstats 4.2 software; GDP, P, and Inv were from statistical year book. 

PGDP reflects the level of social and economic development and the degree of affluence. P reflects population size. UC reflects the CEs per unit GDP, and indirectly reflects the change of technology level. These indices are of great socioeconomic significance and are the conventional driving factors widely used in current research. GI refers to the output value of unit fixed assets investment, which reflects the transformation ability of investment to economic output. High GI value and high investment conversion rate. UInv refers to the investment in fixed assets per unit GDP output, which reflects the intensity of social and economic expansion of reproduction. Since fixed asset investment (Inv) and economic output (GDP) can be regarded as input and output, respectively, they reflect the driving pattern of economic growth. The smaller UInv value, the lower the investment intensity, the higher the investment utilization rate, the higher the technical content, as the technology drives the economic growth. The higher UInv value, the greater the investment intensity, the lower the investment utilization rate, the lower the technical content, as the investment drives the economic growth. Few studies have applied UInv to the study of LMDI model [38]. SPLIT equals the total landscape area squared divided by the sum of patch area squared, summed across all patches of the corresponding patch type. SPLIT refers to the level of looseness of landscape spatial distribution. The smaller the value, the looser the spatial form is. On the contrary, the higher the level of aggregation is. MESH is constrained by the ratio of cell size to landscape area and is achieved when the corresponding patch type consists of a single pixel patch. The larger the value, the larger the average patch area, the larger the total area. ED refers to the ratio of perimeter to area, which reflects the increase of boundary length and the expansion of area. The higher the value, the greater the level of fragmentation. TE is the reciprocal of the circumference, which is used to measure the expansion of the area. The smaller the value is, the more significant the expansion is, and the total area increases. In the existing literature, few studies have applied SPLIT, MESH and ED to Satial-LMDI model. These three indices reflect the change of spatial form of built-up areas. The introduction of these three factors overcomes the shortcomings of the existing LMDI model, because the existing LMDI model is unable to detect the impact of spatial morphology changes on CEs. Therefore, the Spatial-LMDI model proposed in this study allows us to explore the driving factors of CEs changes from the spatial and socio-economic levels.

Taking logarithms on both sides of the formula and referring to the addition Spatial-LMDI decomposition rules adopted by [39,40], we rewrite the Formula (2) as follows:(3)$$\Delta C=\Delta P+\Delta PGDP+\Delta UInv+\Delta UC+\Delta GI+\Delta SPLIT+\Delta MESH+\Delta ED+\Delta \left(TE\right)$$
(4)$$\Delta C=C_{t+1}-C_{t}$$
(5)$$\Delta P=\frac{C_{t+1}-C_{t}}{lnC_{t+1}∆lnC_{t}}\times \ln\;\left(\frac{P_{t+1}}{P_{t}}\right)$$
(6)$$\Delta PGDP=\frac{C_{t+1}-C_{t}}{lnC_{t+1}-lnC_{t}}\times \ln\;\left(\frac{PGDP_{t+1}}{PGDP_{t}}\right)$$
(7)$$\Delta UInv=\frac{C_{t+1}∆C_{t}}{lnC_{t+1}-lnC_{t}}\times \ln\;\left(\frac{UInv_{t+1}}{UInv_{t}}\right)$$
(8)$$\Delta UC=\frac{C_{t+1}-C_{t}}{lnC_{t+1}-lnC_{t}}\times \ln\;\left(\frac{UC_{t+1}}{UC_{t}}\right)$$
(9)$$\Delta GI=\frac{C_{t+1}-C_{t}}{lnC_{t+1}-lnC_{t}}\times \ln\;\left(\frac{GI_{t+1}}{GI_{t}}\right)$$
(10)$$\Delta SPLIT=\frac{C_{t+1}-C_{t}}{lnC_{t+1}-lnC_{t}}\times \ln\;\left(\frac{SPLIT_{t+1}}{SPLIT_{t}}\right)$$
(11)$$\Delta MESH=\frac{C_{t+1}-C_{t}}{lnC_{t+1}-lnC_{t}}\times \ln\;\left(\frac{MESH_{t+1}}{MESH_{t}}\right)$$
(12)$$\Delta ED=\frac{C_{t+1}-C_{t}}{lnC_{t+1}-lnC_{t}}\times \ln\;\left(\frac{ED_{t+1}}{ED_{t}}\right)$$
(13)$$\Delta TE=\frac{C_{t+1}-C_{t}}{lnC_{t+1}-lnC_{t}}\times \ln\;\left(\frac{TE_{t+1}}{TE_{t}}\right)$$

Therefore, the final driving factors of CEs change are divided into seven effects, include nine factors. They are population effect (population size Δ*P* and *P*), economic effect (economic level Δ*PGDP* and *GDP*), intensity effect (investment intensity *UInv* and Δ*UInv*), technology effect (technology level Δ*UC* and *UC*), and efficiency effect (investment efficiency Δ*GI* and *GI*). Aggregation effect (separation degree Δ*SPLIT* and *SPLIT*, edge density Δ*ED* and *ED*), expansion effect (Effective mesh size Δ*MESH* and *MESH*, perimeter index *TE* and Δ*TE*). Negative values for these factors indicate negative effects, and positive values indicate positive effects.

## 3. Results and Discussion

### 3.1. Carbon Dioxide Emission Accounting

In 2019, China’s total CEs in five sectors amounted to 11.96 billion tons, an increase of 297.44% compared with the amount in 1995, with an average annual growth rate (AAGR) of 5.92% (Figure 1a). Among them, Industry CEs increased from 2.30 billion tons in 1995 to 8.85 billion tons in 2019, with an AAGR of 5.78%. Residential CEs increased 322.55%, with an AAGR of 6.19%. CEs from the Traffic, construction and agriculture sectors increased with AAGRs of 8.80%, 7.54% and 2.40%, respectively (Figure 1a). From the decomposition of CEs, the proportion of Industry decreased from 76.42% in 1995 to 74.03% in 2019, which shows that industrial CEs are still a major component of China’s CEs (Figure 1b). In addition, the proportion of household CEs decreased from 13.68% to 14.54%, while that of Traffic, Construction and Agriculture are 7.77%, 1.49% and 2.17% in 2019, respectively (Figure 1b).

The growth of CEs in China can be divided into three main stages: the low growth stage in 1995–2000, the fast growth stage in 2000–2012 and the deceleration stage in 2012–2019. From 1995 to 2000, China was in the early stage of the Reform and Opening-up [41], and the economy was in a period of low-speed growth, with an AAGR of 0.97% in GDP and an AAGR of 2.22% in CEs. Since China joined the World Trade Organization (WTO) in 2001, China’s economy entered the fast lane [42]. From 2000 to 2012, the AAGR of China’s GDP is 3.58%, and the AAGR of CEs is 10.56% in a period of rapid growth. After 2012, due to the economic restructuring and upgrading in China [43] and the outbreak of the global financial crisis in 2008 [44], economic growth slowed down. On the other hand, industrial emissions in China slowed down due to the industrial transformation and upgrading. The AAGR of GDP decreased to 1.07%, while the AAGR of CEs decreased significantly to 3.27%. It could be seen that the changing trend of social economy is consistent with that of CEs.

In order to understand the relationship between CEs and socio-economic changes, we explored the relationship between per capita CEs, per capita industry CEs, per capita household CEs, per capita building CEs and per capita Agriculture CEs, and per capita GDP CEs and per capita GDP (Figure 2). Among them, GDP is the quantitative indicator for comparison. In addition, per capita GDP is used to indicate the level of social affluence and economic development level. Per capita residential CEs and per capita Agriculture CEs are used to indicate the per capita energy consumption level of residents and agricultural activities. Per capita Industrial CEs and per capita construction CEs are used to represent the per capita level of CEs of industry and construction, indirectly reflecting the level of social and economic development. The results show that per capita CEs increase from 3.04 tons per person to 8.51 tons per person with the growth of per capita GDP, with a growth rate of 243.84 %. This reflects that the improvement of social affluence has a positive impact on CEs (Figure 2). Among them, the per capita industry CEs increased 233.06% with the growth of per capita GDP, which indicates that the improvement of social affluence promotes the consumption of energy and industrial products, this leading to the increase of industrial CEs. The growth trend of industrial CEs is consistent with per capita CEs, because industrial CEs are the main component of CEs, accounting for more than 70% of total CEs. In addition, per capita residential CEs, per capita traffic CEs, per capita Agriculture CEs and per capita construction CEs increased by 265.57%, 554.60%, 56.10% and 395.73%, respectively. This shows that the level of social affluence has a positive effect on the living standards of residents and economic development. Interestingly, the per unit GDP CEs increased from 0.63 ton/thousand yuan to 1.25 ton/thousand yuan. Per unit GDP CEs has three increase rates, the first stage is the annual increase rate −0.34% with the per capita GDP from 4.72 to 4.93 thousand yuan per person. The second stage has the highest increase rate of 11.78%, with the per capita GDP increased from 5.03 to 6.25 thousand yuan per person. During the latest low increase stage, annual increase rate slowed down to 0.61%. This three stages reflect the continuous maturity and advancement of China’s industrial structure, due to the economic development, industrial upgrading and societal increasing awareness of energy conservation and emission reduction. 

With respect to the changes of CEs in terms of provinces distribution, Shandong Province experienced the greatest increase in CEs (959.40 million tons), followed by Jiangsu (736.82 million tons), Guangdong (660.37 million tons), Hebei (654.96 million tons) and Zhejiang (491.33 million tons) between 1995 and 2019 (Figure 3). From 1995 to 2019, Hainan had the lowest increment in CEs (44.30 million tons), and followed by Beijing (48.10 million tons). Ningxia has the highest AAGR (10.47%), which indicated that the growth rate of CEs in Ningxia Province was higher than that in other provinces, and followed by Hainan (10.42%), Inner Mongolia (10.31%). It is notable that Hainan has high AAGR and low increase, which means Hainan’s economy developed fast, and demanded rapidly growing energy relatively, but its absolute increase amount is smaller than other provinces. Two of three provinces with the largest AAGR are located in western China. From 1995 to 2019, Beijing’s GDP increased 152.50%, with CEs growth of 63.29%, and the ratio of CEs growth rate to GDP growth rate is 0.52, remaining the lowest in China. This shows that there is a weak decoupling relationship between economic growth and CEs growth in Beijing, that is, economic growth is less dependent on energy consumption, so its economic development quality is higher. One of the main reasons for this phenomenon is that the industrial restructuring in Beijing has led to the gradual upgrading of energy-intensive industries and the transfer to other provinces [45,46]. In terms of spatial distribution, most of the provinces with a large increase in CEs, they located in the southeastern coast and central regions of China, while the provinces have a fast growth rate are mostly located in the western and eastern regions. The southeastern coastal and central regions of China are important areas of China’s economy and the most populous areas (accounting for 65.02% of the national population). Therefore, there is a huge demand for energy in the southeastern coastal and central regions, and the increment of CEs is larger than west region. The highest growth rate of CEs in the western region (such as Ningxia) is due to the implementation of the strategy of developing the western region, which has enabled the western economy to develop and the investment to increase gradually, so the demand for energy and CEs are growing rapidly. The three provinces with the largest ratio of CE growth rate to GDP growth rate are Inner Mongolia (7.45), Heilongjiang (7.05) and Hainan (4.56), GDP growth is highly dependent on energy consumption.

### 3.2. Analysis of Driving Force

The change of CEs in China has experienced three stages: low growth period (1995–2000), accelerated growth period (2000–2012) and medium growth period (2012–2019) (Figure 1a). From 1995 to 2019, P, PGDP, UInv and ED were positive factors for the increase of CEs (Figure 4). SPLIT, GI, UC and TE played a positive role in reducing CEs. In addition, MESH has a positive value on the increase of CEs from 1995 to 2019, which played a positive effect from 1995 to 2019 (Figure 4). This means that the expansion of built-up areas in the context of urbanization will promote the increase of CEs. SPLIT is a positive factor in the low growth period (1995–2000), while it is a negative factor in 2000–2019. This is due to the low level of urbanization and the loose spatial pattern of cities in China during the period of low-speed growth. Therefore, the correlation between cities is weak, and the development of cities is still in an independent stage of development, which is not conducive to reducing CEs. However, due to the acceleration of urban expansion and the increasingly close relationship between cities (such as the rise of urban agglomerations), the spatial distribution between cities becomes more and more concentrated, and the population is also more concentrated, which is conducive to reducing transportation energy consumption and improving the utilization rate of infrastructure, so industrial division of labor and cooperation is more reasonable and efficient, thus reducing CEs.

The contribution of each factor to the increase of CEs is listed in Table 2. From 1995 to 2019, the positive factors contributing to the increase of CEs were PGDP (2363.47%), MESH (14,568.32%), UInv (19,371.61%), ED (2025.84%) and P (939.25%). At the same time, the positive factors for reducing CEs are UC (−13,727.04%), SPLIT (−14,565.66%), GI (−19,371.61%), and TE (−2028.79%). As a result, CEs increased by 297.44% between 1995 and 2019. The greatest positive driving force for the growth of CEs is UInv, which shows that the investment intensity is the main driving force for the increase of CEs, UInv and CEs are strongly coupled (Table 2). 

### 3.3. Regional Differences

In order to understand the different regional driving forces of CEs in China, we accounted for the four major regions (East, Central, Western and Northeastern, Figure 5) [47]. On this basis, the driving factors of CEs change in different regions were analyzed (Figure 5). CEs in the Eastern region increased the most (4.16 billion tons). The Western (2.66 billion tons), the Central region (1.61 billion tons) and the Northeastern (0.52 billion tons) followed. The fastest AAGR of CEs was the Western (6.71%), followed by the Eastern and Central (6.60%), and the lowest AAGR was found in the Northeastern (3.36%) (Figure 5). The implementation of China’s western development strategy and industrial transfer policy has promoted the industrial and social economic development of the west, which has led to the transfer of low-end industries from the Eastern China to Western China [48,49]. Therefore, the Western region is in the downstream of the industrial chain, which leads to high CEs. Northeastern China, as the historical heavy industry base in China, was highly affected by the industrial transformation and upgrading in recent years, for example, the transformation of resource-based cities [50]. Industrial transfer refers to the major changes in the spatial distribution of industries, mainly due to the level of economic development and factor endowments, which lead to the process of repositioning some industrial locations. China’s industrial transfer is mainly about switching backward and energy-intensive industries to the central and western regions, while the eastern region of China focus on developing high-tech and high value-added industries [51]. Due to China’s industrial transfer policy, China’s industrial layout and economic development have undergone significant changes [52,53]. Accordingly, the economic growth rate in Northeastern China slowed down, with the average economic growth rate in 2008–2015 decreased by 18.12% compared with 1995–2008, which led to a significant reduction in the growth rate of heavy industry CEs, and consequently led to a decrease in CEs (28.34%) [54,55]. From the result we conclude that China’s energy resources are concentrated in the central and western regions, and the eastern coastal region is the most economically developed region and the largest energy consumer in China. Therefore, industrial layout, energy endowment and economic development policies jointly affect the regional energy consumption structure and carbon emissions. Some studies have reported the influencing factors of carbon emissions from energy consumption in different regions of China, they point out that technological level is an important factor affecting the structure of regional energy consumption and determines the differences in regional carbon emissions [56].

## 4. Conclusions

Understanding and revealing the driving force of urban CEs change, as well as the relationship between CEs and urban form, were important prerequisites for mitigating global climate change and achieving the CEs reduction goals of the Chinese government. In this study, a bottom-up accounting method was adopted to calculate urban energy consumption CEs at provincial level in China. Landscape index (SPLIT, MESH, ED, TE) was creatively applied to Spatial-LMDI decomposition analysis, then we further explored the relationship between CEs and urban form. This study provided an effective reference for reducing CEs and formulating energy-saving and CEs reduction policies. The results of this study showed that during 2011–2019, China’s CEs increasing rate was slower than GDP, and a relative decoupling relationship between CEs and GDP was found. In addition, there were significant differences in the economic growth and growth rate of CEs among provinces. Spatial-LMDI decomposition analysis shows that the main driving forces of CEs change in five periods were different. The intensity effect, expansion effect and economic effect had different impact on CEs increasement. UC, SPLIT and UInv were important factors to restrain the growth of CEs. Moreover, the increment of CEs in Eastern China was the largest, while the growth rate in Western China was the fastest. The Spatial-LMDI decomposition analysis showed that UInv was the primary driving factor for the growth of CEs, followed by PGDP and P. GI, SPLIT, UC were the three restraining forces to CEs increase, with their unique contributions to the change of CEs among five periods. From the analysis results, the LMDI method mainly focuses on the base and the reporting year, while the interpretation of the annual changes during the study period is unclear. We will explore the annual changes in following researches, after we get access to continuous time series of spatial data.

The policy recommendations based on the above conclusions include the following points: Firstly, the improvement of economic level and social affluence will inevitably lead to the increase of energy consumption, therefore improving energy efficiency and technology level are the ways to help improve the quality of economic development and alleviate the rapid increase of CEs. Secondly, the expansion effect contributes to the increase of CEs, while the aggregation helps to restrain the growth of CEs. Therefore, improving the compactness of urban spatial distribution and controlling the unregulated expansion of construction land are the proper policies to reduce CEs and save land resources. Thirdly, investment intensity has a positive impact on the growth of CEs. Therefore, rational use of investment and improvement of investment efficiency and input–output ratio should be adopted to reducing CEs. Meanwhile, the ecological environment and improving the quality of economic development should be improved accordingly. The Chinese government has taken positive actions for carbon emission reduction, including: (1) Promote new urbanization, by transferring rural population to cities and towns, while saving land resources and facilitating centralized energy supply. This is beneficial to improve energy utilization efficiency and reduce the loss of energy supply [57]; (2) strive to achieve carbon neutrality goals, and put forward energy consumption regulation requirements for provincial governments and industrial departments, such as the dual energy consumption control requirements proposed by the Chinese government (http://www.gov.cn/zhengce/2021-09/18/content_5638215.htm, accessed on 10 January 2021); (3) vigorously develop the information industry and provide policy support for the development of the information industry [58]; and (4) develop and use clean energy, promote electric vehicles, etc. [59]. In addition, the construction of smart cities is also conducive to reducing carbon emissions and formulating policies on reduction of carbon emissions. The improvement of regional collaborative innovation capabilities will also help to improve economic efficiency and energy efficiency [60]. With the support from multiple sensors and big data centers, smart cities can help decision makers rationally dispatch energy and make decisions, thereby reducing energy consumption and carbon emissions.

## Figures and Tables

**Figure 1 ijerph-19-02976-f001:**
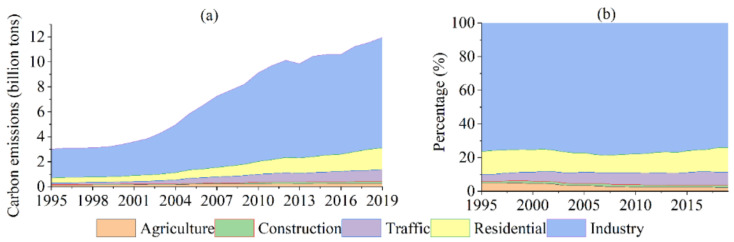
Trends in CEs from 1995 to 2019. (**a**) carbon emissions of China from 1995 to 2019; (**b**) carbon emission percentages of five sectors of China.

**Figure 2 ijerph-19-02976-f002:**
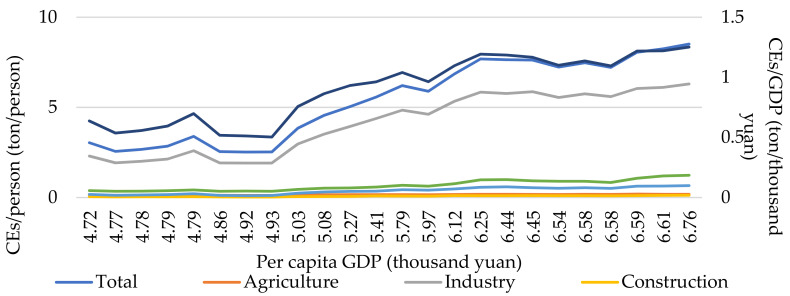
Per capita CEs and per GDP CEs versus per capita GDP.

**Figure 3 ijerph-19-02976-f003:**
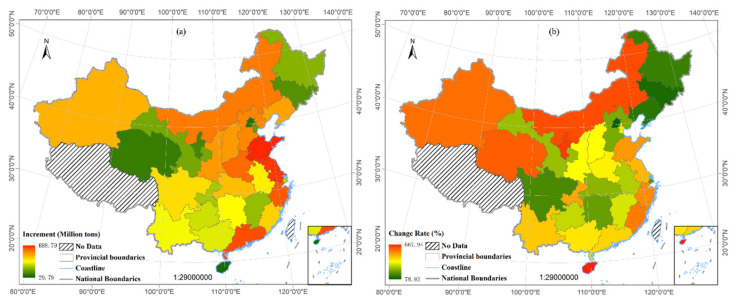
CEs increase (**a**) and AAGR (**b**) of 30 Provinces, China.

**Figure 4 ijerph-19-02976-f004:**
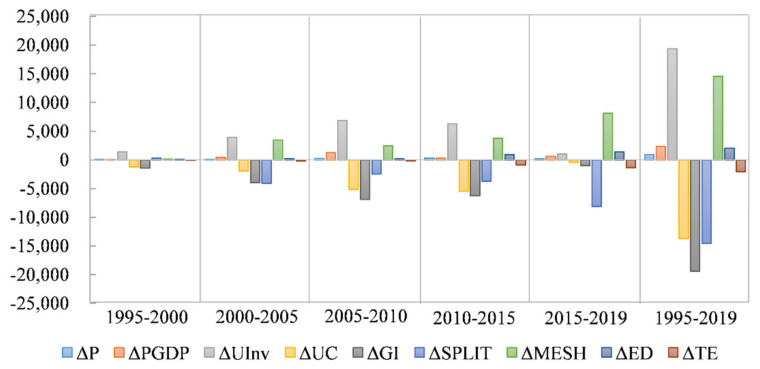
Factorization of four stages; the number is in percentage indicating the growth rate of each component.

**Figure 5 ijerph-19-02976-f005:**
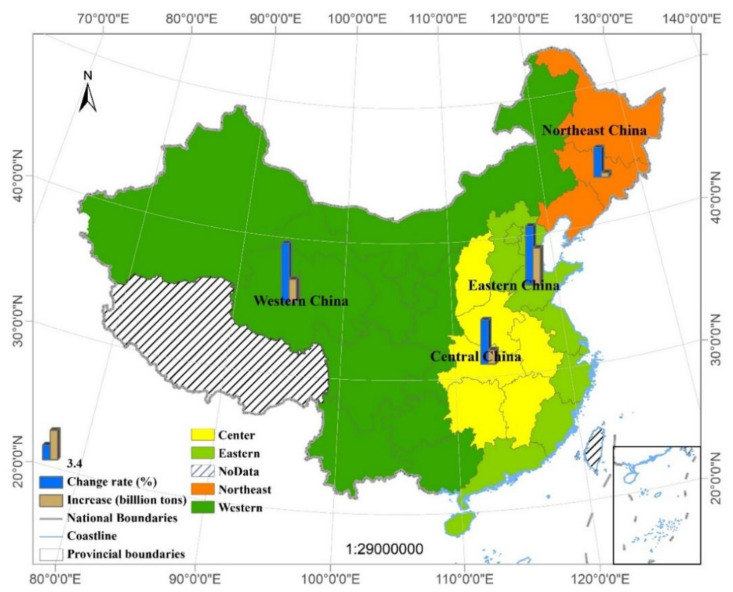
Four research areas in China and their corresponding CEs increments and growth rates from 1995 to 2019. The eastern part includes 11 provinces and municipalities directly under the Central Government (Beijing, Tianjin, Shanghai, Hebei, Jiangsu, Zhejiang, Fujian, Shandong, Guangdong, Hainan, and Taiwan (lack of data). The central part includes six provincial administrative units in Shanxi, Anhui, Jiangxi, Henan, Hubei and Hunan. The West includes 12 provinces, autonomous regions and municipalities directly under the Central Government (Inner Mongolia Autonomous Region, Guangxi Zhuang Autonomous Region, Chongqing City, Sichuan Province, Guizhou Province, Yunnan Province, Tibet Autonomous Region, Shaanxi Province, Gansu Province, Qinghai Province, Ningxia Hui Autonomous Region and Tibet Autonomous Region (lack of data).

**Table 1 ijerph-19-02976-t001:** Decomposition of CEs changes in existing literature.

Scale	Sectors	Decomposition Factors	Study Area & Time	Source
Global	Electricity	Geographical shift, energy mix, share and efficiency, emission factor	Global, 1990–2013	[19]
	Industry	Carbon intensity, energy structure, industrial energy intensity, economic structure, economic development, and population, respectively	Global, 1990–2017	[20]
	Electricity	Socioeconomic indicators	Global, 1990–2014	[21]
	Export	the aggregate carbon intensity, the aggregate weight	Global, 2014	[22]
	Energy-related	Emission factor, energy structure and intensity, income, population	Global, 1980–2015	[23]
Country	Cement	Activity, cement structure, electricity intensity, emission factors	China, 1990–2009	[24]
	Industry	Carbon intensity, energy mix and intensity, industrial activity, employment	China, 1991–2010	[25]
	Coal	Economic scale, industrial structure, energy intensity and mix	China, 1997–2014	[26]
	Energy-related	Population, income, energy intensity, energy structure, and carbon intensity	China and US, 2000–2014	[27]
	Chemistry industry	Carbon intensity, energy structure and intensity, output per worker, economic scale	China, 1981–2011	[28]
	Power	Carbon intensity, energy efficiency and density, economic scale, population	Pakistan	[29]
	Industry	Carbon intensity, energy intensity and structure	China, 1996–2012	[30]
Region	Seven sectors	Socioeconomic and energy	Region, 1996–2012	[31]
	Electricity	Carbon density, energy structure, energy intensity, industrial structure, economic intensity	Latin America & Caribbean, 1990–2015	[32]
	Industry	Emission intensity, energy structure and intensity, economic structure and output, population	Beijing-Tianjin-Hebei, 1996–2000	[33]
Province	Electricity	Electric production, electricity structure, energy efficiency, energy mix, emission factor	Shandong, 1995–2012	[34]
	Energy-related	Energy mix and intensity, economic activity, labor, investment	Liaoning, 1995–2012	[35]
	Energy-related	Population, economic output, energy intensity and energy mix	30 Provinces of China, 2005–2011	[36]
City	Industry	Industrial structure, economic growth and industrial structure	9 cities in Pearl River Delta, 2006–2014	[37]
	Energy-related	Energy structure, energy intensity, industrial structure, population density, and area of construction land	Shanghai, 1999–2015	[17]

**Table 2 ijerph-19-02976-t002:** The contribution of each factor to the change of CEs in five periods.

Factors	1995–2000	2000–2005	2005–2010	2010–2015	2015–2019	1995–2019
ΔP	108.86	91.41	268.32	340.15	216.65	939.25
ΔPGDP	44.19	435.23	1252.60	296.69	601.20	2363.47
ΔUInv	1428.24	3926.06	6890.04	6283.41	1006.58	19,371.61
ΔUC	−1232.12	−1961.43	−5135.41	−5456.78	−456.57	−13,727.04
ΔGI	−1428.24	−3926.06	−6890.04	−6283.41	−1006.58	−19,371.61
ΔSPLIT	313.99	−4079.95	−2477.14	−3734.09	−8107.53	−14,565.66
ΔMESH	131.84	3451.62	2477.91	3735.28	8107.01	14,568.32
ΔED	75.96	202.43	187.90	915.83	1405.29	2025.84
ΔTE	−76.63	−202.76	−189.22	−916.88	−1404.00	−2028.79
ΔC	11.61	74.20	56.01	16.04	12.92	297.44

Note: the number is in percentage indicating the growth rate of each component. Negative figures indicate a positive contribution to reducing CEs.

## Data Availability

Not applicable.

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
