# Peer review of "Quantifying the Impact of Urban Form and Socio-Economic Development on China’s Carbon Emissions"

_ijerph, 2022, doi:10.3390/ijerph19052976_

Round 1
Reviewer 1 Report
The LMDI approach has been widely used in the field of energy and environmental research but the addition of geospatial information in this study offers some unique data points. However, LMDI focuses primarily on the base and reporting year. For the time period of the study, 1995 - 2020 it is not clear how the research method accounts for the annual changes during the research period. Please address this in the discussion.
Author Response
Response to reviewer 1: Thank you for your comments. As you mentioned, the LMDI method mainly focuses on the base and the reporting year, but might not clearly explain the annual variation during the study period. Additional time series data can better explain the characteristics of annual changes, but we lack the access to the continuous time series of spatial data. Therefore we cannot use time series analysis methods, and it is also why we chose to using the LMDI method in this study. We addressed this issue in the discussion. (Page14 Line315-317)Reviewer 2 Report
It would be interesting to correlate the results with pandemic crisis and mention how the COVID19 affected all this phenomenon and with the economic shocks caused.
Author Response
Thank you for your suggestion. Due to space limitations and topic focuses, we will conduct a detailed analysis about the correlation between COVID19 and the results of this study in subsequent studies.
Reviewer 3 Report
It is a good work!
It just seems to us that the notion of Lifestyle should appear in the text and that extractive activities related to smart city development should also appear in the production of greenhouse gases.
Author Response
Thank you for your good suggestion. We discussed the impact of lifestyle on carbon emissions in the revised manuscript, and in future researches we will explore the contribution of smart cities and technologies to reducing urban greenhouse gas emissions. (Page14 Line334-337)
Reviewer 4 Report
The major concern:
The manuscript missing any discussion with the existing literature. This must be improved. The references concentrate mainly on papers investigating China situation. The international perspective should be presented as well – both in the literature review as well as in the discussion section.
Remaining comments:
- Please avoid repetitions, e.g. lines 22-23: “Global warming is one of the most serious threats mankind face and carbon dioxide is a major greenhouse gas causing 22 global warming (Pachauri et al., 2014).”
- The table 1 content or the title should be improved. It focuses primarily on China, while the title suggests general (worldwide) literature review. No study on North America, Africa or Europe is included.
- Section: “2.1 Study area and materials”. Please add the information about the period, for which data were collected. The description includes only land use data details.
- Formula 2 and lines 113-119 – please amend discrepancies in uppercase/lowercase letters in the formula and in the text below. Also: repetitions in description of variables should be excluded (e.g. P, GI).
- 1: please explain why “2019” in the title is in red. Additionally, please add the explanation why 2020 is not taken into account (data not available?, compare the comment on section 2.1).
- 2. The X axis is not readable
- Lines 230-231: Please explain why the names of two periods are identical? If they are the same, why distinguish between the former and the latter?: “accelerated growth period (2000-2012) and accelerated growth period (2012-2019)”
- 4: what are the units of changes showed in the figure?
- Table 2: the title is not clear (at least): g. in what way “high energy” contributes to changes in the population (first line).
- Please check formatting requirements of the journal.
- Language check is recommended.
Author Response
Thank you for your comments. we responsed all quastions in the attachment.

Reviewer 5 Report
The journal (IJERPH) particularly has a specific formatting for the manuscript so I would like to encourage the authors to format the manuscript with the consistent style, not having a different format for main text, tables, figures and references. Table 1 should be organized to enhance the readability. It would be very recommendable to separate Fig 1 and Fig 3 into two figures, respectively.
Table 1 well presents the previous papers that used the LMDI method. It would be very recommendable to add Hoekstra et al. (2003), which illustrates that the index decomposition analysis (IDA) is more flexible, only needs the aggregate data. (Hoekstra, R., Bergh van den, J.C.J.M. 2003. Comparing structural decomposition analysis and sector-level index number analysis. Energy Economics, 2003, 25:39-64.)
[This is important] You should provide the detailed definitions and measurements of GI, UInv, SPLIT, MESH, ED and TE. What does “Splitting index” mean? Please provide a table (or tables) that explain(s) how you measured or calculated those variables.
The 3.1 section is about the current situation on CEs in China. Would it be better if you reorganize the paper by putting “3.1 Carbon Dioxide Emission Accounting” after Introduction?
In Table 2, ΔSPLIT is positive in the period of 1995-2000. You mentioned the SPLIT factor is irrelevant to the increase in CEs for the reason. The SPLIT factor, however, is actually positive. What does the positive sign mean? Similarly, why is the sign of ΔMESH negative in 1995-2000?
The section of Regional Differences is very weak. It is not well capitulated the ideas and objectives that authors seek. First of all, you should provide some information on the current status about the contributing factors on CEs for each region (for example, where are industrial facilities concentrated on?). Please explain more on the “industrial transfer polity”. In addition, the Regional Differences section seems nothing to do with the spatial-LMDI. What is the purpose and status of that section?
The policy recommendations are too generic. What are the current conditions and policy practices? What are the regional policy tools? How do we differentiate the strategies of alleviation by region? Please provide more on the “spatial” dimensions in order the decrease CEs.
Author Response
Thank you for your comments, we responsed all quastions in the attachment.

Round 2
Reviewer 4 Report
The revised version of the manuscript provides a fairly good scienfitic work. All my comments and suggestions were taken into account by the Authors. Although a more extensive discussion with the literature could be added in the conclusions (I advice to back more strongly future reserach in this area by showing more countries with varied background on the regional or local level - this wider perspective would add to the discussion and conclusions for your country), this version is good enough to be published.
Reviewer 5 Report
The authors satisfactorily responded to each question I have raised. The current version of the maunscript is good to be published in IJERPH.